

# Assessment of genetic diversity by phenological traits, field performance, and Start Codon Targeted (SCoT) polymorphism marker of seventeen soybean genotypes (*Glycine max* L.)

Mahmoud Abdel-Sattar[1], Ehab M. Zayed[2], Mohamed K. Abou-Shlell[3], Hail Z. Rihan[4], Ahmed A. Helal[5], Nabil E.G. Mekhaile[6] and Ghada E. El-Badan[7]

[1] Department of Plant Production, College of Food and Agriculture Sciences, King Saud University, Riyadh, Saudi Arabia
[2] Cell Study Research Department, Field Crops Research Institute, Agriculture Research Center, Giza, Egypt
[3] Department of Agricultural Botany (General Botany), Faculty of Agriculture, Al-Azhar University (Assiut Branch), Assiut, Egypt
[4] School of Biological and Marine Sciences, University of Plymouth, Plymouth, United Kingdom
[5] Genetic Resources Research Department, Field Crops Research Institute, Agriculture Research Center, Giza, Egypt
[6] Central Laboratory for Design & Statistical Analysis Research, Agricultural Research Center, Giza, Egypt
[7] Botany and Microbiology Department, Faculty of Science, Alexandria University, Alexandria, Egypt

Corresponding author
Mahmoud Abdel-Sattar,
mmarzouk1@ksu.edu.sa

## ABSTRACT

The Egyptian-farmed soybeans have a wide range of genetic diversity which is most important in plant improvement programs in order to develop new higher yielding soybean genotypes. The present study is designed to determine the genetic variability among seventeen genotypes of cultivated soybean (*Glycine max* L.) by examining the phenotypic level at the seedling stage, field performance over two years 2022/2023 and genetically using Start Codon Targeted (SCoT) markers. Results indicated that the SCoT markers, 100 seed weight, and tip angle (TA) traits were positively correlated with H2L12, DR 101, H15L5, and H117 genotypes. In addition, the number of branches per plant and plant height were associated with H113, H32, Crowford, H129, and D7512035. Furthermore, the length of the first internode (LFI), root width (RW), root length (RL), and shoot length (SL) were more associated with Giza 111, NC105, and Hutcheson. The hierarchical cluster analysis (HCA) and its associated heatmap explored the differences among the genotypes. It showed that all examined parameters were clustered into four distinct clusters. The obtained results showed that genotypes NC105, H30, D75_12035, and H2L12 have promising phenological and morphological traits besides tracking the inheritance of nearby genes surrounding the ATG translation start codon since they are in a monoclades. The obtained results will help the breeder plan appropriate selection strategies for improving seed yield in soybeans through hybridization from divergent clusters.

## INTRODUCTION

Soybean (*Glycine max* L.) is an important plant legume for feed and food items, given its high nutritional protein and oil resource, amino acids, vitamins, minerals, and other nutrients, it is low cost with excellent functional properties for farm households with limited resources in rural economies (*Ashry et al., 2018*; *Zamaisya & Nyikahadol, 2018*; *Han et al., 2024*). The crop has a higher percentage of protein than any other crop legume, (30–45%) protein and (15–24%) beneficial saturated fats with a low percentage of monounsaturated fats (*El-Hashash, 2016*; *Dilawari et al., 2022*). Therefore, it is widely used in the human diet, oil industry, food industry, and livestock feed industry particularly for chickens, as an important food ingredient and functional additive (*Akharume, Aluko & Adedeji, 2021*; *Zhang et al., 2024*).

The soybean species (2n = 40) belong to the family Fabaceae and the Egyptian farmed soybeans have a wide range of maturity and diverse morphology (*Metwally et al., 2018*; *Shilpashree et al., 2021*). Sufficient genetic information regarding soybean traits is essential in soybean breeding programs by introducing well-adapted varieties or through hybridization and selection for one or more of the major yield components. Furthermore, knowledge of genetic variability is most important in plant improvement programs in order to develop new higher-yielding soybean genotypes. The assessment of the crop plant's genetic diversity as a whole is one of the requirements for effective breeding techniques. The foundation for creating an effective breeding program is the availability of genetic diversity among plant materials. Identification of the type and extent of genetic variety aids in the selection of varied parents for intentional hybridization by the plant breeder (*Govindaraj, Vetriventhan & Srinivasan, 2015*; *Ikegaya, Shirasawa & Fujino, 2023*). Morphological criteria might not be sufficient to distinguish between types of soybeans with a limited genetic basis. Furthermore, many studies have been conducted to understand the scope and genetic bases of variability across major characteristics in soybean genotypes (*Morsy et al., 2011*; *Shilpashree et al., 2021*).

Different morphological features like flower, pubescence, seed, and hilum color, and physiological and biochemical traits such as protein content, oil, carbohydrates, and their subcomponents to explore the genetic variation in soybeans were employed by *Boerma & Specht (2004)*; *Jin et al. (2023)*; *Vera, Priano & Vázquez (2024)*. In addition, agronomic, morphological, biochemical, and molecular marker polymorphisms have all been utilized by *Morsy et al. (2011)*; *Goyal, Sharma & Gill (2012)*; *Zatybekov et al. (2023)*. All of the aforementioned marker groups when used collectively, can deliver accurate data regarding the tested germplasm (*Sudaric et al., 2008*; *Kujane, Sedibe & Mofokeng, 2021*).

In order to identify and evaluate the evolutionary links among soybean cultivars, morphological characteristics, molecular, and biochemical analyses have been carried out for many years (*Shilpashree et al., 2021*; *Khan et al., 2023*). Molecular markers are crucial for addressing the evolutionary relationships between and among the different species and cultivars. Several molecular markers have been developed to identify cultivars and study evolutionary relationships between different genomes to explore genetic diversity (*Bornet & Branchard, 2004*; *Semagn, Bjornstad & Ndjiondjop, 2006*; *Zhang et al., 2021*). *Collard &*

*Mackill (2009)* developed the SCoT polymorphism as a unique, dominant, rapid, and creative DNA marker. The flanking short section of the translation initiation or the conserved short-start codon ATG seen in plant genes are the targets of this marker. In comparison to previous DNA markers like random amplified polymorphic DNA (RAPD), it was characterized by assessing kinship relationships. Compared to RAPDs, ISSRs, and SSRs, it is efficiently utilized for the development of marker-assisted breeding techniques (*Mulpuri, Muddanuru & Francis, 2013*; *Vanijajiva, 2020*). In numerous crop plant species, the SCoT markers have been applied, such as rice (*Collard & Mackill, 2009*; *Patidar et al., 2022*), cowpea (*Igwe et al., 2017*; *Hussein & Osman, 2020*), Iranian Plantago (*Rahimi et al., 2018*) and wheat (*Nosair, 2020*; *Abouseada et al., 2023*).

Soybean breeders work hard to gain proper knowledge of the extent and genetic basis of variability throughout crucial characteristics in soybean genotypes. As a result, the current study aimed to assess the genetic variation in seventeen soybean genotypes that have not been investigated before, using phenological traits, field behavior, and SCoT polymorphism marker. Accordingly, correlation and cluster analysis were applied to ascertain the evolutionary connections between genotypes, seed yield, and their pertinent characteristics. This will also help gather data on the extent and nature of genetic components of variation dominating the expression of yield and the associated attributes in these soybean genotypes analyzed to improve productivity.

## MATERIALS & METHODS

### Plant materials

Seventeen Egyptian soybean (*Glycine max* L.) genotypes were kindly supplied from the Genetic Resources Research Department, Field Crops Research Institute, Agricultural Research Center (ARC), Giza, Egypt. The pedigree and origin of genotypes were taken from *Akram et al. (2011)*; *Morsy, Mohamed & Abou-Sin (2016)*; *Guo et al. (2022)* and coded (Table 1). Planted place in Spring, 15th of May which is the soybean growing season, as soybean planting begins in May in Egypt.

### Determination of phenological characteristics

The current experiment was carried out in the laboratory of the Agricultural Botany Department, Faculty of Agriculture, Al-Azhar University Assiut branch in the 2023 season to examine the phenological characteristics of soybean plant seedlings. The seeds of seventeen soybean genotypes were planted in 14 cm diameter plastic pots that were filled with 2 kg of a mixture of clay and sand (1:2 w: w), with 10–15 seeds in each pot. The seventeen cultivars in this experiment were divided into three replications (three pots for each replicate) Different phenological characteristics of plant samples were taken 26 days after sowing. Three plants per replicate from each genotype were randomly taken. The plants were separated into their organs (roots, stems, and leaves). Scan images from different genotypes were utilized to get quantitative measurements of the phenological traits as follows: Root traits (root number (RN), root maximum length (RL), root width (RW) and root tip angle (TA°)); Shoot traits (shoot length (SL), shoot length of first internodes (LFI) and shoot diameter of first internodes (DFI)); Leaf traits (leaf area (LA),

**Table 1** The studied soybean genotypes and their codes.

| Code No. | Genotype | Pedigree | Origin | Code No. | Genotype | Pedigree | Origin |
|---|---|---|---|---|---|---|---|
| 1 | NC105 | D55-4110 x N56–4071 | Egypt | 10 | Toano | AES, USA[**] | America |
| 2 | H117 | FCRI[*] | Egypt | 11 | Holladay | AES, USA[**] | America |
| 3 | H113 | FCRI[*] | Egypt | 12 | DR101 | AES, USA[**] | America |
| 4 | H129 | FCRI[*] | Egypt | 13 | Giza35 | FCRI[*] | Egypt |
| 5 | NC104 | D75-4110 x N56-4071 | America | 14 | Giza111 | FCRI[*] | Egypt |
| 6 | H30 | FCRI[*] | Egypt | 15 | Crawford | USA[***] | America |
| 7 | H32 | FCRI[*] | Egypt | 16 | D75_12035 | Govan x F4 line (Bragg x PI 229358) | America |
| 8 | H2L12 | FCRI[*] | Egypt | 17 | Hutcheson | V68-1034 (York x PI 71506) x Essex | America |
| 9 | H15L5 | FCRI[*] | Egypt | | | | |

Notes.
[*]FCRI, Field Crops Research Institute, Giza, Egypt.
[**]AES, USA, Agricultural Experiment Station, USA.
[***]USA, US Regional Soybean Laboratory at Urbana, Illinois, and Stoneville, Mississippi.

leaf length (LL) and leaf width (LW)) and Seed traits (single seed weight (SW), seed length (SL), seed area (SA) and seed weight (SW)).

## Field experiments

The current experiment was conducted over two years 2022/2023 at Bahtim Research Station, Agricultural Research Center, Kaliobia Governorate, Egypt (Latitude 30°8′22″N, Longitude N 31°15′50″) to examine the field performance of soybean crops. The seeds of seventeen soybean genotypes were planted after where one waits until the soil is firm and can withstand the feet, and then the row is opened and planted. Planting was in rows in three replicates, each replicate consisting of three rows each 3 m, and the distance between the rows is 70 in a randomized complete block design (RCBD), and the distance between the holes is 15 cm. The soybean genotypes were planted at a density of 30 plants/m in a single row on the row. Soybean genotypes received the standard agricultural practices according to the recommendation of the Ministry of Agriculture, Egypt by instructions agriculture extension were followed regarding fertilization and irrigation until the harvest was complete. The experimental soil texture was clay. The soil physicochemical properties were characterized by ana-lysing samples from 30 cm depth (Table 2). Growth parameters including plant height, number of branches, number of pods, number of seeds, seed yield, and one hundred seed weight were measured.

## Molecular evaluations

To examine the SCoT polymorphism marker, approximately 500 mg of young and fresh leaves from each genotype of five-week-old plants of seventeen soybean genotypes were collected from the first experiment. Genomic DNA was extracted from fresh plant leaves by the DNeasy plant mini kit 69204 (Bio Basic, Amherst, MA, USA). To assess DNA purity, the ratio of absorbance at 260 and 280 nm is used using a UV spectrophotometer. The consensus sequence was used to design ten SCoT primers from *Joshi et al. (1997)*; *Collard*

**Table 2  Physical and chemical analyses of the experimental soil.**

| Soil depth (cm) | Soil fractions | | | Soil texture | pH (1:2.5)[*] | EC (dS/m) | CaCO₃ (%) | OM (%) | Available elements mg kg⁻¹ | | | |
|---|---|---|---|---|---|---|---|---|---|---|---|---|
| | Sand (%) | Clay (%) | Silt (%) | | | | | | | | | |
| 0–30 | 21.60 | 52.74 | 25.66 | Clay | 7.85 | 0.57 | 2.50 | | N | P | K | Fe |
| | Soluble Cations (meq/L) | | | | Soluble Anions (meq/L) | | | 1.30 | 40.68 | 4.20 | 220.00 | 5.71 |
| Soluble cations and anions | Na⁺ | Ca⁺² | Mg⁺² | K⁺ | HCO₃⁻ | Cl⁻ | SO₄⁻ | | Mn | Zn | Cu | Pb |
| | 1.27 | 2.10 | 1.96 | 0.32 | 1.84 | 1.49 | 2.32 | | 4.32 | 3.18 | 3.39 | 0.54 |

**Notes.**
*in soil and water suspension.

**Table 3  Primer sequences used in the SCoT analysis.**

| Primer name | Sequences (5′–3′) | % GC |
|---|---|---|
| SCoT 2 | ACCATGGCTACCACCGGC | 67 |
| SCoT 3 | ACGACATGGCGACCCACA | 61 |
| SCoT 4 | ACCATGGCTACCACCGCA | 61 |
| SCoT 6 | CAATGGCTACCACTA CAG | 50 |
| SCoT 9 | ACAATGGCTACCACTGCC | 56 |
| SCoT 10 | ACAATGGCTACCACCAGC | 56 |
| SCoT 11 | ACAATGGCTACCACTACC | 50 |
| SCoT 13 | ACCATGGCTACCACGGCA | 61 |

*& Mackill (2009)*; *Mohamed, Shoaib & Gadalla (2015)*; *Alotaibi & Abd-Elgawad (2022)*. All SCoT primers were 18-mer and were from Dataset I, which is based on highly expressed genes as described by *Sawant et al. (1999)*. Eight primers were selected, which gave very noticeable and consistent bands for the data's final amplification (Table 3). Amplification reactions were performed following (*Xiong et al., 2011*; *Fathi, Hussein & Mohamed, 2013*; *Alotaibi & Abd-Elgawad, 2022*) where, an annealing at 50 °C for 50 s. and a prolongation stage at 60 s. The preliminary expansion section was expanded to 7 min at 72 °C within the last cycle and was carried out in the Techni®TC-512 Thermal Cycler (UK). Amplified PCR fragments were electrophoresed on 1.5% agarose gel with staining dye ethidium bromide to visualize DNA and a 1kb ladder marker (Bio-Rad, Hercules, CA, USA). The runs were carried out for exactly 30 min at 100 V in a mini-submarine gel (Bio-Rad).

## Statistical analysis

All phenotypic data were evaluated for normal significance using the Shapiro–Wilk test ($\alpha = 0.05$), the standard deviation was calculated, and a one-way ANOVA using SPSS ver. 22.0 and a graphical representation was created using a JMP® ver.16 cell plot (*SAS Institute Inc, 2008* software version 9.13). To compare a genotype's performance, PLABSTAT software (*Utz, 2001*) used genotypes and the three replications as fixed and random effects. The following parameters were calculated with a 5% criterion of significance for all tests: Heritability (H2) = Genotypic variance (2G)/ Phenotypic variance (2p); Genetic coefficient

of variation (GCV) and Correlation coefficients for each trait using PLABSTAT software's GENOT function.

For phenotypic data, a constellation plot of the hierarchical clustering dendrogram was produced using JMP® ver.16 and Ward's technique. For field data, clustering of genotypes was performed based on Ward linkage and Euclidean distances as an "r" matrix (*Everitt, 1993*; *Eisen et al., 1998*).

DNA banding patterns were photographed using the Bio-1D gel documentation system. All gels were analyzed by Gel Analyzer 3 software, which scored clear amplicons as present (1) or absent (0) for each primer and entered them in the form of a binary data matrix. The following are several descriptive measures of diversity: Band's number (total, monomorphic, polymorphic, and unique); percent of polymorphism (Pb %) and Marker efficiency indices (Heterozygosity index (H), Polymorphic Information Content (PIC), effective multiplex ratio (E), Arithmetic mean of H (H.av), Marker In-dex (MI) and discriminating power (D) according to *Chňapek et al. (2024)*, calculated by the Marker Efficiency Calculator (*iMEC, 2018*)). The Dice coefficient was used to assess the genetic similarity of genotypes using a (0/1) data matrix (*Adhikari et al., 2015*; *Khattab et al., 2022*). Heatmap was performed by using the R software through the web tool ClustVis (*Adhikari et al., 2015*) to visualize similarities and dissimilarities among genotypes. Collectively, ClustVis, a web tool for visualizing the clustering of multivariate data was used to illustrate and calculate relationships, heatmap, and genetic trees based on penology, field performance traits, and SCoT according to *Metsalu & Vilo (2015)*.

# RESULTS

## Phenological traits

In this study, fourteen quantitative traits for the seventeen soybean genotypes were examined. The mean root parameters and shoot stem parameters of the seventeen soybean genotypes are shown in Table 4. Root parameters and shoot stem parameters showed significant differences among all genotypes ($P < 0.05$). For root parameters (Table 4 and Fig. 1), the root number (RN) of all genotypes was between 6.67 and 26.67 while the shortest (6.67 cm) was recorded in H113. Root maximum length (21.66 cm) (RL) was recorded in H30 genotype while the shortest (6.91 cm) was recorded in H117. The root width (RW) (cm) of Crawford was the highest (0.40 cm) while the lowest (0.25 cm) was recorded in D75_12035. Ttip angle of root T.A of H117 was the longest (64.19°), while, the shortest (26.93°) was recorded in H113. Herein, the Crawford genotype had the highest root width value (0.40 cm) and the lowest (0.25 cm) was found in the D75_12035 genotype. Results illustrated that the largest root tip acute angle (TA) was in the H117 genotype (64.185°) and the smallest was recorded in the H113 genotype (26.928°). Additionally, in the H113 genotype, a wider root system (RW) was linked to a reduced degree of tip angle for seminal roots. For shoot stem parameters, the significant differences revealed that there is significant variability among the soybean genotypes for all the characteristics investigated at 0.05 level of probability (Table 4 and Fig. 1), shoot length (SL) (cm) of NC105 was the longest (21.83 cm), while, the shortest (9.10 cm) was recorded in H117. The length of the first internodes

**Table 4  Root and Stem morphometric traits for 26-day seedling.**

| Code no. | Genotypes | Root parameters | | | | Shoot stem parameters | | |
|---|---|---|---|---|---|---|---|---|
| | | RN[1] | RL[2] (cm) | RW[3] (cm) | TA[4] (°) | SL[5] (cm) | LFI[6] (cm) | DFI[7] (cm) |
| 1 | NC105 | 13.67def | 20.50ab | 0.36ab | 48.07abcd | 21.83a | 7.65a | 0.24bcde |
| 2 | H117 | 11.00fg | 6.91d | 0.27bc | 64.19a | 9.10 h | 4.96d | 0.17f |
| 3 | H113 | 6.67 h | 16.73abc | 0.34abc | 26.93d | 13.49defg | 6.35abcd | 0.22cdef |
| 4 | H129 | 8.67gh | 15.02bc | 0.33abc | 38.20cd | 13.40defg | 5.41bcd | 0.28abc |
| 5 | NC104 | 17.67c | 16.20abc | 0.32abc | 56.00abc | 12.34fg | 6.51abc | 0.30ab |
| 6 | H30 | 11.00fg | 21.66a | 0.36ab | 51.05abc | 17.01bc | 6.66abc | 0.23cdef |
| 7 | H32 | 15.00cde | 18.49ab | 0.34abc | 41.65bcd | 17.20 bc | 5.62bcd | 0.21def |
| 8 | H2L12 | 21.00b | 15.69bc | 0.34abc | 57.55abc | 11.64gh | 5.38bcd | 0.28abc |
| 9 | H15L5 | 12.67ef | 18.55ab | 0.28bc | 41.88bcd | 13.17efg | 5.70bcd | 0.28abc |
| 10 | Toano | 21.00b | 14.86bc | 0.36ab | 49.46abc | 11.82fgh | 6.85ab | 0.33a |
| 11 | Holladay | 14.00def | 17.07abc | 0.32abc | 48.69abcd | 12.24fgh | 5.16cd | 0.24bcd |
| 12 | DR101 | 16.67cd | 12.04cd | 0.40a | 62.80 ab | 14.93cdef | 5.60bcd | 0.18ef |
| 13 | Giza35 | 22.00b | 16.41abc | 0.26bc | 44.00abcd | 16.20cde | 5.31cd | 0.25bcd |
| 14 | Giza111 | 26.67a | 18.87 ab | 0.39a | 43.25abcd | 16.44cd | 6.47abcd | 0.25bcd |
| 15 | Crawford | 12.67ef | 17.81 ab | 0.40a | 48.67abcd | 19.74ab | 5.99bcd | 0.23cdef |
| 16 | D75_12035 | 8.00gh | 16.17abc | 0.25c | 51.27abc | 17.19bc | 6.20abcd | 0.25bcd |
| 17 | Hutcheson | 24.00ab | 18.29ab | 0.34abc | 37.11cd | 17.14bc | 6.07bcd | 0.30ab |
| | LSD 0.05 | 3.25 | 5.70 | 0.10 | 21.78 | 3.20 | 1.53 | 0.06 |

**Notes.**
Data represents means of three replicates ± standard deviation (SD).
[1] Root number.
[2] Root maximum length
[3] Root width
[4] Tip angle of root
[5] Shoot length
[6] Length of first internodes
[7] Diameter of first internode
Mean values within a column for each season followed by different letters are significantly different at $P \leq 0.05$.

(LFI) values was between 4.96 and 7.65 cm. The diameter of the first internode (DFI) values ranged from 0.17–0.33 cm.

The mean values for the measured and calculated leaf parameters and seed parameters of the seventeen soybean genotypes are shown in Table 5. All genotypes demonstrated substantial differences in the leaf parameters and seed parameters phenotypic traits ($P < 0.05$). Phenological screening showed that the three essential leaf traits' range: area (LA), length (LL), and width (LW) were in the ranges of 6.442–13.884 cm2, 2.603–4.256 cm, and 2.756–4.967 cm and, respectively. The data revealed that the widest leaf area was from in NC105 genotype while the smallest one for in the Toano genotype, respectively. Collectively, the Toano genotype had the lowest LL, LW, and LA values, while the NC105 genotype had the highest SL, LFI, and LA values, and also and H113 genotype had the maximum LL and LW values. Analysis of seed parameters of soybean genotypes are shown in Table 5 clearing the significant differences between means ($P < 0.05$). Single seed weight, seed length, seed area, and seed weight of the 17 soybean genotypes ranged from 7.21 to
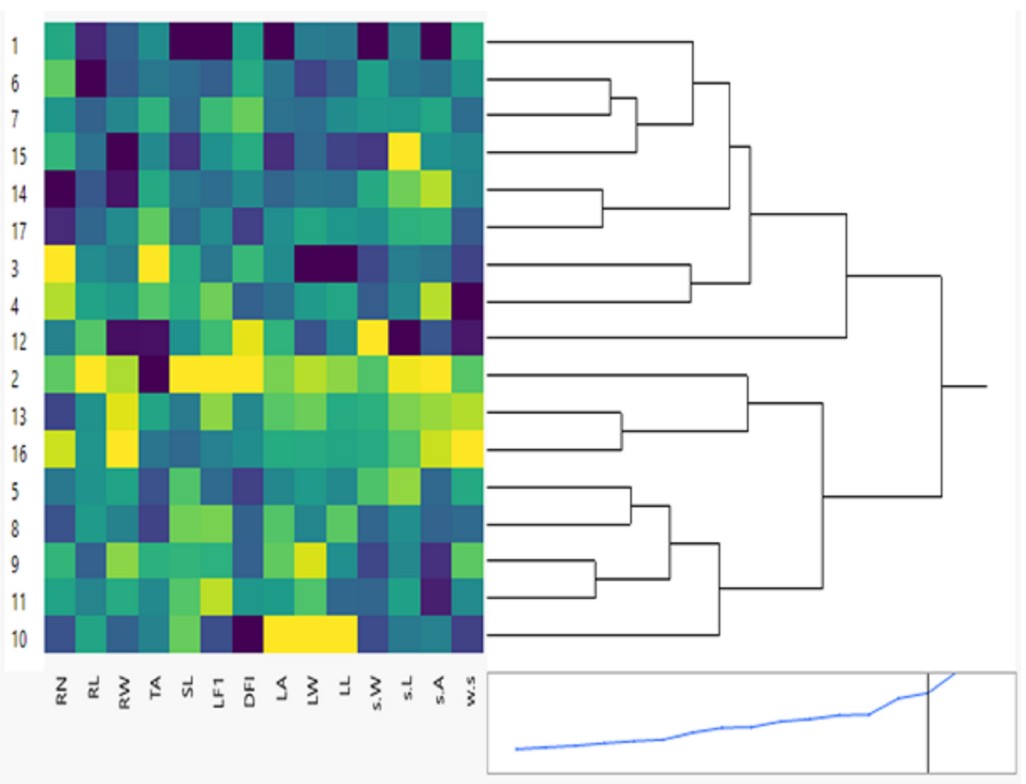

**Figure 1  Clustering multivariate analysis of seventeen soybean genotypes for phenological and morphological traits using hierarchical co-clustering dendrogram and heatmap by Ward's method.** Row clusters were obtained at genotype level; 1. NC105, 2. H117, 3. H113, 4. H129, 5. NC104, 6. H30, 7. H32, 8. H2L12, 9. H15L5, 10. Toano, 11. Holladay, 12. DR101, 13. Giza35, 14. Giza111, 15. Crawford, 16. D75_12035 and 17. Hutcheson. Column clusters were recorded at the trait level.

9.29 9, 6.71 to 9.01 cm, 43.25 to 56.49 cm, and 0.28 to 0.61, respectively. All genotypes demonstrated substantial differences in the root phenotypic traits. In this study, modern automated and semi-automated methods of geometric shape analysis of phenotypic and geometric traits depart from the coordinates' points for seed parameters. These traits are single seed weight (SW), seed length (SL), seed area (SA), and weight of seeds (WS). The genotypes NC105, DR101, and H129 had the highest values, whereas the genotypes DR101, Crawford, H117, and D75_12035 had the lowest values (Fig. 1 and Table 5).

The results of phenological traits showed tremendous variation among 26-day stage (26 days) soybean genotypes. These variations are essential for developing seventeen new cultivars with distinct phenological traits. The clustering between phenological and morphological characteristics is shown as a double dendrogram or colored heatmap (Fig. 1). The 17 genotypes were classified into two main clusters. Genotypes: NC105, H113, H129, H30, H32, DR101, Giza111, Crawford, Hutcheson belong to one group but H30 & H32, H113 & H129, Giza111 & Hutcheson more correlated with each other. The other group included H117, NC104, H2L12, H15L5, Toano, Holladay, and Giza35 D75_12035. The intensity of colors reflects visually the high, intermediate, and low characteristic values.

**Table 5 Descriptive statistics of 26-day seedling for leaf morphometric and seed geometric traits.**

| Code no. | Genotypes | Leaf Parameters | | | Seed parameters | | | |
|---|---|---|---|---|---|---|---|---|
| | | LA[1] (cm²) | LW[2] (cm) | LL[3] (cm) | SW[4] (g) | SL[5] (cm) | SA[6] (cm²) | SW[7] (g) |
| 1 | NC105 | 13.88a | 3.47bcd | 4.10bc | 9.29a | 7.54bc | 56.49a | 0.42fg |
| 2 | H117 | 7.77ef | 2.76def | 3.14ef | 7.91 g | 6.74de | 43.25 h | 0.37gh |
| 3 | H113 | 9.86bcde | 4.26a | 4.97a | 8.95bc | 7.61bc | 51.31bcde | 0.55abc |
| 4 | H129 | 10.84bcd | 3.25bcdef | 3.67cde | 8.82bc | 7.45bcd | 44.62gh | 0.61a |
| 5 | NC104 | 10.05bcde | 3.25bcdef | 3.93bcd | 7.92 g | 6.92cde | 51.90bcde | 0.43efg |
| 6 | H30 | 10.65bcde | 3.87ab | 4.28abc | 8.33def | 7.66b | 51.50bcde | 0.46defg |
| 7 | H32 | 10.79bcd | 3.58abc | 3.91bcd | 8.40de | 7.31bcde | 48.42defg | 0.51bcdef |
| 8 | H2L12 | 8.21cdef | 3.40bcde | 3.32def | 8.77c | 7.36bcde | 52.16abcde | 0.51bcde |
| 9 | H15L5 | 8.05def | 2.68ef | 3.89bcde | 8.95bc | 7.45bcd | 54.58abc | 0.37gh |
| 10 | Toano | 6.44f | 2.60 f | 2.76f | 8.92bc | 7.65b | 50.55cde | 0.56abc |
| 11 | Holladay | 9.41cdef | 3.00cdef | 4.24abc | 8.77c | 7.26bcde | 55.31ab | 0.47cdef |
| 12 | DR101 | 8.75cdef | 3.77ab | 3.90bcde | 7.21 h | 9.01a | 52.82abcd | 0.59ab |
| 13 | Giza35 | 8.16def | 2.92cdef | 3.62cde | 8.13fg | 6.96bcde | 45.19fgh | 0.32hi |
| 14 | Giza111 | 11.20abc | 3.52abc | 4.15bc | 8.21ef | 6.99bcde | 44.65gh | 0.48cdef |
| 15 | Crawford | 12.80ab | 3.62abc | 4.53ab | 9.02b | 6.71e | 49.60def | 0.47cdef |
| 16 | D75_12035 | 8.95cdef | 3.14bcfef | 3.66cde | 8.19ef | 7.05bcde | 44.29gh | 0.28i |
| 17 | Hutcheson | 9.80bcde | 3.18bcdef | 3.81bcde | 8.48d | 7.17bcde | 47.73efgh | 0.53abcd |
| | LSD 0.05 | 3.00 | 0.76 | 0.77 | 0.24 | 0.73 | 4.58 | 0.09 |

**Notes.**

Data represents means of three replicates ± standard deviation (SD).

[1] Leaf area
[2] Leaf width
[3] Leaf length
[4] Single seed weight
[5] Seeds length
[6] Seed area
[7] Seed weight

Mean values within a column for each season followed by different letters are significantly different at $P \leq 0.05$.

Dark colors like blue usually indicate higher data values for genotypes NC105, H113, H30, and Crawford. Cooler colors like green and yellow represent lower data values which mostly for second group genotype Toano in leaf traits.

## Field performance

Through the data across the two seasons, the data was collected and a homogeneity of variance analysis was performed in the 2022/2023 seasons *via* Bartlett's test. For this, pooled variance across the two seasons was used. The combined analysis of variance after the homogeneity test for error variances, the *F*-test, and the mean performances of the seventeen soybean genotypes for different traits are performed (Table 6). The test revealed that there were no significant differences in the effect of the year for all traits, while the effect of genotypes was significant for all traits except for 100-seed weight. Analysis of the field behavior of the seventeen soybean genotypes are shown in Table 6. There were significant differences between all the field performance parameters of the 17 soybean genotypes (*P*

**Table 6  Mean performance of seventeen soybean genotypes for the studied traits.** The values show combined data over two seasons. Values within a column for each season that are followed by different letters are significantly different at $P \leq 0.05$.

| Code no. | Genotype | Traits | | | | | |
|---|---|---|---|---|---|---|---|
| | | Plant height (cm) | No. of branches per plant | No. of Pods per plant | No. of seeds per plant | Seed yield per plant (g) | 100-seed weight (g) |
| 1 | NC105 | 39.63j | 3.88abc | 75.00a | 175.40bc | 27.34bcde | 15.56 |
| 2 | H117 | 66.00e | 4.50a | 50.38 g | 122.00 g | 20.38gh | 17.44 |
| 3 | H113 | 77.13c | 3.75bcd | 35.63i | 88.63i | 15.76i | 17.45 |
| 4 | H129 | 80.88b | 3.38cd | 47.75 g | 135.80ef | 23.23efg | 17.23 |
| 5 | NC104 | 35.88k | 3.13d | 70.63b | 204.00a | 35.42a | 17.67 |
| 6 | H30 | 36.38k | 2.38e | 65.75cd | 188.30b | 31.42ab | 16.67 |
| 7 | H32 | 95.00a | 3.50bcd | 37.88i | 108.10 h | 16.23hi | 15.34 |
| 8 | H2L12 | 56.13f | 3.75bcd | 35.63i | 99.63hi | 16.63hi | 16.52 |
| 9 | H15L5 | 44.00i | 3.63bcd | 48.63 g | 137.60o | 24.15defg | 16.86 |
| 10 | Toano | 43.63i | 3.25cd | 58.25e | 160.40d | 26.94bcde | 16.80 |
| 11 | Holladay | 34.13k | 3.63bcd | 68.50bc | 185.80b | 29.33bc | 15.76 |
| 12 | DR101 | 43.88i | 3.13d | 48.25 g | 128.30efg | 21.50fg | 16.69 |
| 13 | Giza35 | 47.13 h | 3.88abc | 59.25e | 158.80d | 25.80cdef | 15.38 |
| 14 | Giza111 | 48.38 h | 3.50bcd | 74.38a | 202.50a | 29.41bc | 15.69 |
| 15 | Crawford | 52.25 g | 4.13ab | 44.00 h | 124.40fg | 20.26ghi | 16.51 |
| 16 | D75_12035 | 72.75d | 3.75bcd | 54.38f | 141.50e | 24.56defg | 17.16 |
| 17 | Hutcheson | 58.38f | 3.75bcd | 64.00d | 170.60cd | 28.18bcd | 16.44 |
| | LSD 0.05 | 2.29 | 0.40 | 2.98 | 13.54 | 4.584 | NS |

< 0.05). The significant differences revealed that there is a significant variability among the genotypes for all the characteristics investigated. The H32 genotype was the tallest plant (95.0 cm), while had small pod numbers (37.88). The H113 genotype had the lowest number of seeds per plant (88.63) with the lightest seed yield (15.76 g). The H30 genotype was a short plant (36.38 cm) with the fewest number of branches per plant (2.375 cm). Although the NC104 genotype was a small plant (35.88 cm), it had the largest number of seeds per plant (204.0). The Giza111 genotype had the largest pod numbers per plant (74.38) and the most number of seeds per plant (202.5). Other genotypes showed different parameters: Holladay genotype had short plants (34.13 cm), H117 genotype had the most profuse plants (4.5 cm), N105 genotypes had the largest pod numbers per plant (75.0), H2L12 genotypes had the small pod numbers (35.63) and N104 genotype had the greatest seed yield per plant (35.42 g).

In the present analysis, the similarity levels of the 17 soybean genotypes were assessed based on seed yield and its related traits (Fig. 2). They were classified into six main groups (clusters) with high similarity of over 50% in almost all genotypes. The first cluster contains five genotypes with 67% similarity: NC 105, Holladay, Giza 111, Giza 35, and Hutcheson. The second cluster has four genotypes with 64.8% similarity: H117, H113, H2L12, and Crawford. Third shows the highest similarity% (83.96) between genotypes: H129, D

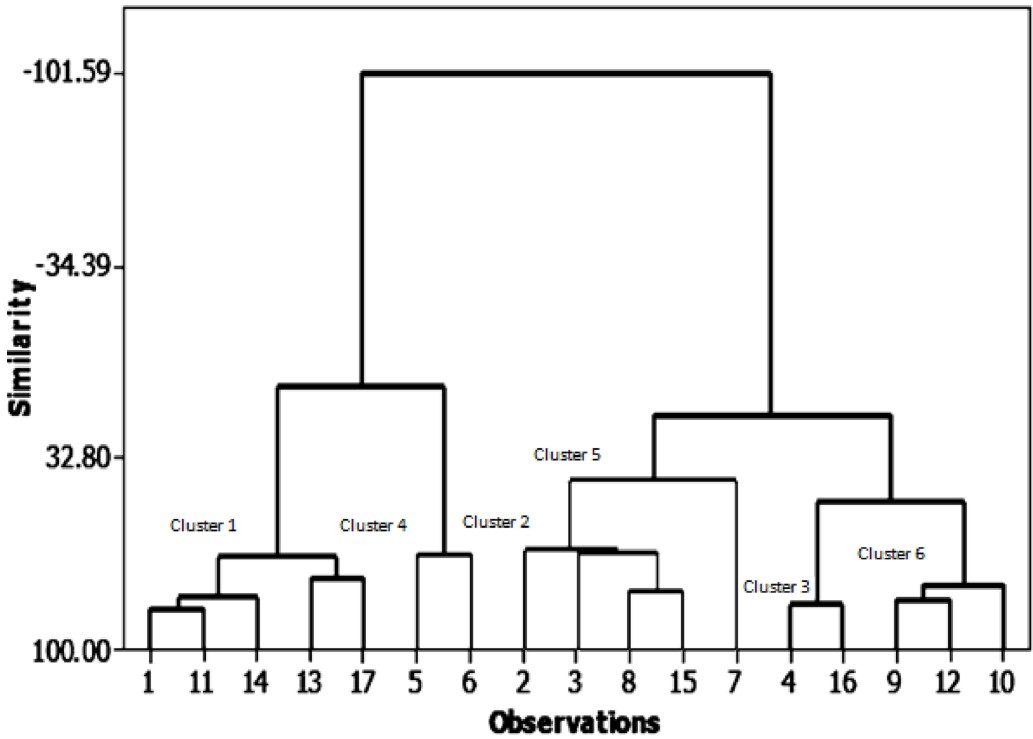

**Figure 2** Cluster analysis of growth parameters for the seventeen soybean genotypes.

75-12035. The fourth cluster encloses NC 104, H 30 geno-types which had the highest pod numbers, seed yield, and 100 seed weight with 66.63% likeness. The fifth clustering includes the H32 genotype that has the highest number of plants and seed number/plant height with 40.67% similarity. Sixth clustering includes H32 and H15L5, DR101, and Toano by 77.42% similarity. The most noticeable relation between phenological characteristics with the field performance is the steady clustering of some genotypes among them than other genotypes: NC105, Giza 111, Hutchesonin & H113, H32, Crawford & H117, H2L12 & H15L5, Toano.

## SCoT polymorphism marker

Out of the 15 primers used in this study, the eight SCoT primers that worked well produced good, repeatable, highly informative, and scorable fingerprint patterns (Fig. 3). Every SCoT primer displayed a distinct banding characteristic with a medium-to-high GC range of 50% to 67%. Out of 20 bands, 12 polymorphic bands were produced by the eight SCoT primers in this instance (Table 7). With an average polymorphism rate of 57%, the percentage of polymorphic bands varied from 50% with SCoT 2 to 75% with SCoT 3. The eight SCoT primers that were selected had varying degrees of effectiveness, which are listed in Table 7. With a mean moderate value of 0.30, the polymorphism information content (PIC) varied from 0.19 for SCoT 10 to 0.46 for SCoT 2. The Heterozygosity Index (H), Arithmetic Mean of H (H.av), Marker Index (MI), and Discriminating Power (D) were all greater for SCoT

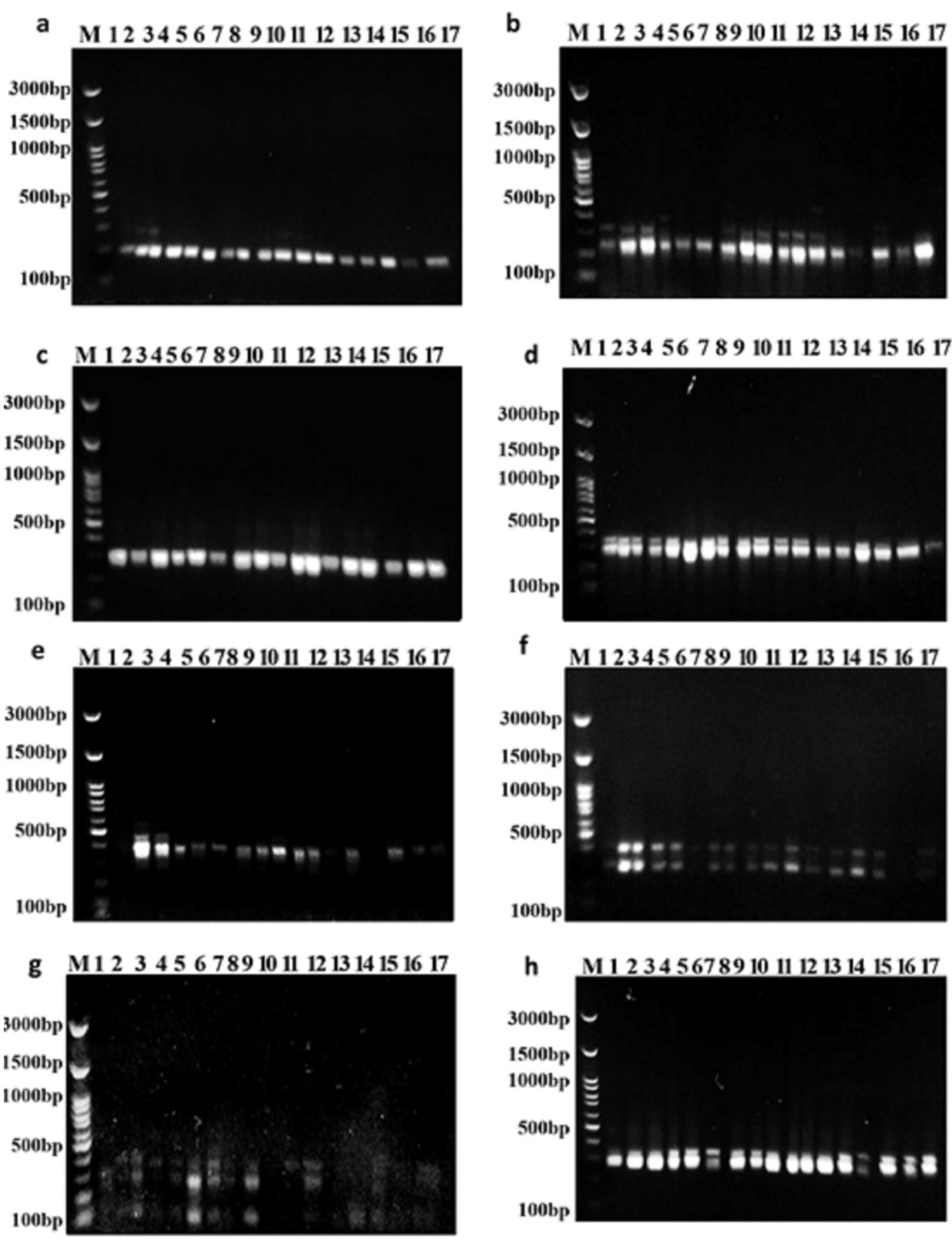

**Figure 3  Amplifications of the seventeen Soybean genotypes using eight SCoT primers.** (A) SCoT 2, (B) SCoT 3, (C) SCoT 4, (D) SCoT 6, (E) SCoT 9, (F) SCoT 10, (G) SCoT 11 H SCoT 13.

3 and SCoT 9. Their mean values range from 0.35 to 0.40, and their values are H: 0.54, 0.52, H.av: 0.54, 0.52, MI: 0.54, 0.52, and D: 0.61, 0.57, respectively.

The data for SCoT is displayed in a heatmap where the grid/row represents a genotype and each column represents a band (Fig. 4). The color and intensity of the boxes are used

Abdel-Sattar et al. (2024), *PeerJ*, DOI 10.7717/peerj.17868

**Table 7** The total number of bands, monomorphic bands, polymorphic bands, unique bands, percentage of polymorphism, and six marker indices of soybean genotypes by eight SCoT primers.

| Primer name | Total band | Monomorphic band | Polymorphic band | Unique band | % of Polymorphic band | Marker indices | | | | | |
|---|---|---|---|---|---|---|---|---|---|---|---|
| | | | | | | H[1] | PIC[2] | E[3] | H.av[4] | MI[5] | D[6] |
| SCoT 2 | 2 | 1 | 1 | 3 | 50.00% | 0.52 | 0.46 | 1.00 | 0.52 | 0.52 | 0.69 |
| SCoT 3 | 4 | 1 | 3 | 2 | 75.00% | **0.54** | **0.43** | 1.00 | **0.54** | **0.54** | **0.61** |
| SCoT 4 | 2 | 1 | 1 | 0 | 50.00% | 0.25 | 0.23 | 1.00 | 0.25 | 0.25 | 0.28 |
| SCoT 6 | 2 | 1 | 1 | 0 | 50.00% | 0.20 | 0.19 | 1.00 | 0.20 | 0.20 | 0.22 |
| SCoT 9 | 3 | 1 | 2 | 0 | 66.66% | **0.52** | **0.42** | **1.00** | **0.52** | **0.52** | **0.57** |
| SCoT 10 | 2 | 1 | 1 | 1 | 50.00% | 0.11 | 0.10 | 1.00 | 0.11 | 0.11 | 0.11 |
| SCoT 11 | 3 | 1 | 2 | 0 | 66.66% | 0.49 | 0.40 | 1.00 | 0.49 | 0.49 | 0.50 |
| SCoT 13 | 2 | 1 | 1 | 0 | 50.00% | 0.20 | 0.19 | 1.00 | 0.20 | 0.20 | 0.22 |
| Total | 20 | 8 | 12 | 3 | | 2.82 | 2.43 | 8.00 | 2.82 | 2.82 | 3.20 |
| Average | 2.50 | 1 | 1.50 | 0.40 | 0.57 | 0.35 | 0.30 | 1.00 | 0.35 | 0.35 | 0.40 |

**Notes.**
[1] Heterozygosity index
[2] Polymorphic Information Content
[3] Effective multiplex ratio
[4] Arithmetic mean of H
[5] Marker Index
[6] Discriminating power
Bold styling indicates high values for each marker index.

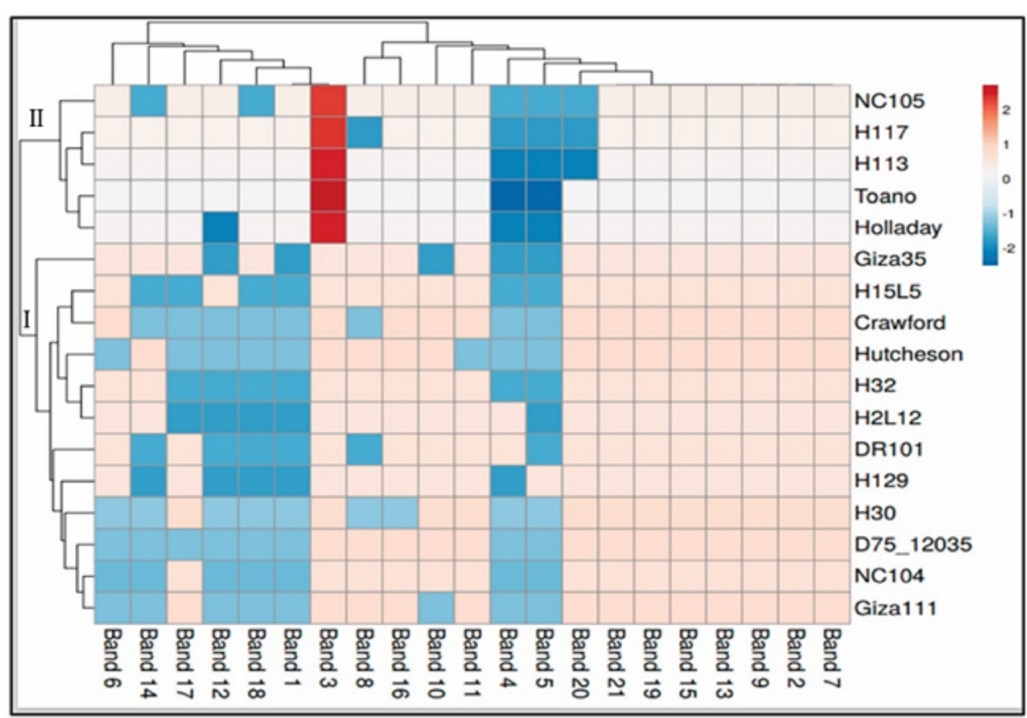

**Figure 4** Hierarchical clustering by heatmap of seventeen soybean genotypes with the bands (gene) regions for SCoTanalysis using Euclidean distance.

to represent changes in similarities in DNA (not absolute values). The color scale from dark red (the least-used parts of a gird) as in five genotypes of cluster II (NC105, H117, H113, Toano, Holladay) to blue (the most popular parts of the gird) for the rest of the genotypes of cluster I. Dark red indicate more activity of genes, while cooler colors indicate less.

## The complete assessment of the phenological traits, field performance, and SCoT polymorphism marker

The hierarchical cluster analysis (HCA) and its associated heatmap explored the differences among the genotypes (Fig. 5). Rows correspond to the studied Soybean genotypes, whereas columns correspond to different phenology, Field parameters, and SCoTs markers. Low numerical values are green colored, while high numerical values are colored rose (see the scale at the bottom left corner of the heat map). Briefly, the HCA-associated dendrogram based on seedling morphological characterization (phenology trait), field performance, and SCoT molecular markers outputs showed that they were clustered separately into four distinct clusters. Cluster 'A' included NC105, Giza 111, Hutcheson, and H30, while cluster 'B' consisted of NC104 Toano, Holiday, and Giza 35. Furthermore, cluster 'C' included H117, H29, H113, H32, D7512035 and H15L5, while H2L12, DR101, and Crawford were located in cluster 'D'. Concerning the heatmap correlation, cluster A &B genotypes had a strong correlation. Moreover, there were moderate correlations among genotypes of cluster C, while the strongest correlations were found between cluster D genotypes.

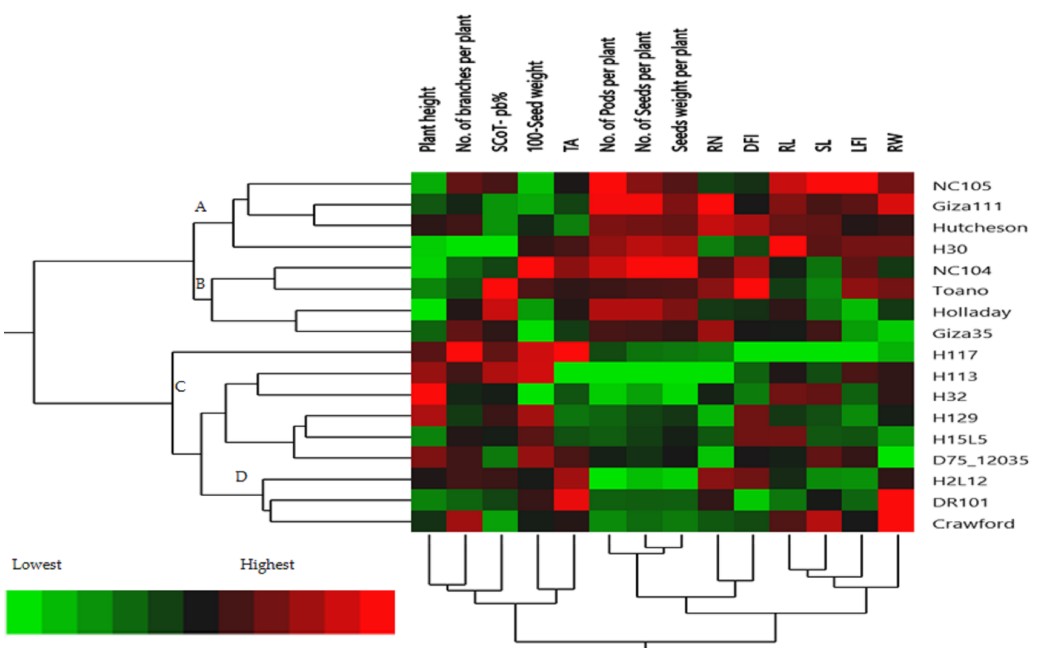

**Figure 5  Two-way hierarchical cluster analysis (HCA) of seventeen soybean genotypes associated with the soybean seedling morphology, field performance and genetic variability.**

## DISCUSSION

Based on diversity in plant genetic resources, the identification of genetic links between plant genotypes provides to development of new and improved cultivars with desirable characteristics and significant insights that simplify the process of selecting farmer-preferred traits such as yield potential and large seed, *etc* (*Hromadová et al., 2023*). In addition, diverse genetic resources provide breeders with preferred traits of germplasm for breeding such as pest and disease resistance and photosensitivity, *etc.*, and facilitate the development of effective conservation measures (*Govindaraj, Vetriventhan & Srinivasan, 2015*). So, this study expansively evaluated field behavior, phenological traits, and SCoT polymorphism markers among seventeen soybean genotypes.

Herein, the Crawford genotype had the highest root width value (0.40 cm) and the lowest (0.25 cm) was found in the D75_12035 genotype. Additionally, in the H113 genotype, a wider root system (RW) was linked to a reduced degree of tip angle for seminal roots. The architectural root traits include the number (RN), length (RL), width (RW), and tip angles (TA) beside the emergence, depth, convex hull area, and root mass center (*Atkinson et al., 2015*; *Pan et al., 2023*). The phasic development of the seedling is also greatly influenced by the architectural features of its roots. Surprisingly, the large root system postponed maturity and as a result, more grain stuffing. This association is most likely the result of improved nutrient and water intake for photosynthesis (*Pinto & Reynolds, 2015*; *Lynch et al., 2023*). A Cell plot of phenological parameters revealed significant differences between seedlings of 16-day-old soybean genotypes at Z1.1 growth stage. Collectively, genotype W2

had the lowest internode length (SL), length of first internode (LFI), and leaf area (LA) values, while genotype W8 had the highest diameter of first internode (DFI) and LA values. Furthermore, the W7 genotype had maximum plant biomass (PB) and leaf width (LW). Our results are also in agreement with those of *El-Esawi et al. (2023)*, who reported that Wheat (*Triticum aestivum* L.) genotypes showed significant genetic variation using a cell plot of phenological parameters. In addition, significant differences among the 16-day-old seedlings of shoot length, length of first internodes, leaf area, diameter of first internode, plant biomass, and leaf width (LW) values.

Direct correlation coefficients among seed yield/plant and the further related traits. Foremost, a positive correlation between seed yield/plant with each of pod numbers/plant, seed number/plant, and between the number of plants bearing pods and plants bearing seeds. On the other hand, there was a highly negative correlation, as mentioned by *Al Barri & Shtaya (2013)*; *Berhanu, Tesso & Lule (2021)*; *Moustafa, Zubaidah & Kuswantoro (2021)*. The similarity levels of genotypes were assessed based on seed yield and its related traits. They were classified into six main groups (clusters). The fifth cluster had one genotype (H32) with the highest number of plants and seed number/plant height. Cluster number four consisted of two genotypes (NC104 and H30), which had the highest pod numbers per plant, seed yield/plant, and 100 seed weight. The similarity level between the genotypes in the fourth cluster was 66.63%. These findings are in agreement with those results obtained by *Shadakshari et al. (2011)* for 50 soybean genotypes concerning seed traits per plant and seed yield per plant, and (*Sheykhi et al., 2014*) showed that the cluster analysis between 30 bread wheat genotypes showed that days to flowering, grain dry weight, stem diameter, panicle dry weight and number of kernels per panicle were the most closely related to grain yield. In the same line, *Pallavi, Jiban & Ujjawal (2020)* explained that 76.6% of total variability among sixteen soybean genotypes is attributable to plant height, days to maturity, number of pods/plant, 100 seed weight, and grain yield.. There was a strong opportunity to acquire enough scope for genotypic improvement in soybeans through hybridization among genotypes picked from any divergent clusters.

Among the cultivated varieties, they identified several highly conserved regions, indicating selection during domestication (*Valliyodan et al., 2021*). Most of this study found advanced breeding lines as well as seven best tests were evaluated in terms of yield and trait ratios by following a balanced group design. It has been found that many genotypes outperform varieties in screening yield and attributable traits. During the same cropping season, one promising entry was evaluated (*Maranna et al., 2021*).

At the level of the genomic DNA, phylogenetic relationships were described using molecular markers (*Zhang et al., 2021*). A conserved region like ATG flanking the translation start codon makes the creation of SCoT markers very simple (*Xiong et al., 2011*; *Rai, 2023*). In *Triticum* L., *Vicia* L., and *Glycine Max* L. respectively, evaluation of SCoT markers has already been developed (*Abouseada et al., 2023*; *Soliman et al., 2023*; *Vivodík et al., 2023*). All SCoT primers had a medium-to-high GC range of 50% to 67% than that mentioned by (*Rayan & Osman, 2019*) and showed a clear banding profile conflicting with *Marilla & Scoles (1996)* about the linkage between the GC content of primer and the clarification of the banding profile. Those markers revealed an overall

average polymorphism percentage of 57%. This percentage is comparable to prior studies in soybeans by *Fahmy & Salama (2002)*; *Satya et al. (2013)*; *Rayan & Osman (2019)*. Therefore, it is suggested that SCoT 3 and SCoT 9 are the most effective primers. Our research is consistent with other studies, the polymorphism was over 50% (*Guo et al., 2022*; *El Framawy, El bakatoushi & Deif, 2016*). This high polymorphism percentage might be attributed to wide genetic diversity and high conservation among the examined soybean genotypes. SCoT markers and the gene/trait defining them can be directly employed in breeding programs.

To determine the level of polymorphism between soybean genotypes, polymorphic information content (PIC) is calculated (*Agarwal et al., 2018*; *Vivodík et al., 2023*). Since the PIC values ranged from low (0.00–0.25), moderate (0.25–0.5) to high (>0.50) indicating low, moderate, and high levels of informativeness and genetic diversity, as mentioned by *Botstein et al. (1980)*; *Vivodík et al. (2023)*. Our soybean genotypes showed a moderate genetic diversity among them with a mean value of 0.30. Interestingly, the PIC index has been extensively in many genetic diversity studies (*Amom et al., 2020*). PIC analysis can be used to evaluate markers so that the most appropriate marker can be selected for genetic mapping and phylogenetic analysis (*Anderson et al., 1993*; *Powell et al., 1996*; *Adly et al., 2023*; *El-Esawi et al., 2023*). A higher PIC value for a marker reinforces a high value for other indices. The mean values of the heterozygosity index (H), arithmetic mean of H (H.av), Marker Index (MI) and discriminating power (D) ranging from 0.35 to 0.40, indicating a moderate level of polymorphism for a better technique used in a given germplasm pool (*Powell et al., 1996*; *Nagaraju et al., 2001*; *Akash et al., 2023*; *Soliman et al., 2023*). Genetic diversity analysis frequently uses multivariate analytical approaches, which simultaneously analyze several measurements for each genotype. The approach that is currently most frequently employed is cluster analysis. Clustering of the dendrogram and heatmap allows genotypes to show the strength of relationships between them.

The SCoT marker is a useful method for the evolutionary ancestry of some cultivars as wheat (*Xiong et al., 2011*), creating a novel fingerprint for plants (*Aboulila & Mansour, 2017*) and for the discrimination and identification of cultivars (*Mohamed et al., 2017*). The results are similar to those obtained by *Abouseada et al. (2023)*; *El-Esawi et al. (2023)*.

The presence of the genotype distributions in the cumulative Heatmap is due to their basis in breeding programs. At the same time, this also appears as the reason for the emergence of Holiday and Toano among NC104 genotypes. H30, NC105, and Hutcheson genotypes were known as the basis for measuring those traits during the period of breeds of these genotypes. These genotypes explain the breed's concept of soybean breeding by mixing old genotypes with imports from the USDA. Additionally, in the heatmap, the degree of colored girds represents the highly correlated characteristics. The Heatmap showed that while SCoT polymorphism, 100 seed weight, and TA trait were positively correlated with H2L12, DR 101, H15L5, and H117, the number of branches per plant, and plant height were associated with H113, H32, Crowford, H129, and D7512035. Furthermore, LFI, RW, RL, and SL were more associated with Giza 111, NC105, and Hutcheson. Furthermore, the HCA-associated dendrogram in this study showed that genotypes were grouped into four distinct clusters based on the phenotypic characteristics of the seedling, field performance,

and molecular marker. Utilizing multi-factor approaches, making catalogue for each genotype having the best agronomic traits can speed up yield improvement. These could help understand and analyze the results of the complex biological questions posed in the world of agriculture and the impact of genetic variation on plant performance and breeding (*Maulana et al., 2023*; *Omar et al., 2023*).

## CONCLUSIONS

This study demonstrated the phenological, field performance, and genetic variation between seventeen soybean genotypes that will be helpful in planning the appropriate selection of superior genotypes for future strategies based on the phenotypic expression used for improving seed yield in soybeans and the performance of the breeding program to improve the important traits. The distinct clustering and genetic variation of the seventeen genotypes provide directed opportunities for breeders to choose from them. So, it is suggested that genotypes: NC105, H30, D75_12035, and H2L12 will be the best selection for accelerating genetic manipulation and shortening the breeding cycles, especially, the NC105 genotype.

### Funding

This research was funded by the Researchers Supporting Project (number: RSPD2024R707), King Saud University, Riyadh, Saudi Arabia. The funders had no role in study design, data collection and analysis, decision to publish, or preparation of the manuscript.

### Grant Disclosures

The following grant information was disclosed by the authors:
Researchers Supporting Project: RSPD2024R707.
King Saud University, Riyadh, Saudi Arabia.

### Competing Interests

The authors declare there are no competing interests.

### Author Contributions

- Mahmoud Abdel-Sattar conceived and designed the experiments, performed the experiments, analyzed the data, prepared figures and/or tables, authored or reviewed drafts of the article, funding acquisition, project administration, and approved the final draft.
- Ehab M. Zayed conceived and designed the experiments, performed the experiments, analyzed the data, prepared figures and/or tables, authored or reviewed drafts of the article, and approved the final draft.
- Mohamed K. Abou-Shlell conceived and designed the experiments, performed the experiments, analyzed the data, prepared figures and/or tables, authored or reviewed drafts of the article, and approved the final draft.

- Hail Z. Rihan conceived and designed the experiments, analyzed the data, authored or reviewed drafts of the article, and approved the final draft.
- Ahmed A. Helal conceived and designed the experiments, analyzed the data, authored or reviewed drafts of the article, and approved the final draft.
- Nabil E.G. Mekhaile conceived and designed the experiments, analyzed the data, authored or reviewed drafts of the article, and approved the final draft.
- Ghada E. El-Badan conceived and designed the experiments, performed the experiments, analyzed the data, prepared figures and/or tables, authored or reviewed drafts of the article, and approved the final draft.

## Data Availability

The raw measurements are available in the Supplementary File.

## Supplemental Information

Supplemental information for this article can be found online at http://dx.doi.org/10.7717/peerj.17868#supplemental-information.

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
