# Peer review of "Assessment of genetic diversity by phenological traits, field performance, and Start Codon Targeted (SCoT) polymorphism marker of seventeen soybean genotypes (Glycine max L.)"

_PeerJ, doi:10.7717/peerj.17868_

## Round 0.1 · original submission · Minor Revisions

In addition to reviewers, here are my suggestions for improving your manuscript:
-Line 65-66: This sentence should be revised as "Furthermore, much study has been conducted to understand the scope and genetic bases of variability across major characteristics in soybean genotypes".
-Line 95: only ScoT. Because you abbreved it before. Go through the entire text and do this for each abbreviation.
-Line 103: Please delete "Three separate experiments on".
-Line 107: "spring" not "summer"
-Line 110: Instead of "The first experiment", write "Determination of phenological characteristics".
-Line 125: Write "Field experiments" instead of "The second experiment".
-Line 142: Replace "The third experiment" with "Molecular evaluations".
-Line 337-342: If this is not your finding, please write stating that. If this is your finding, why did you cite it? This sentence is not understood.
-Line 353: Did these researchers also find the same similarity rate? Did they also use the same genotypes? You should improve the discussion section.
-Line 377: "is calculated" not "was calculated"
-Line 381: "according to Botstein et al., 1980" please delete.
-Line 394-397: delete this paragraph. It's repetitive and irrelevant.
-Line 403-407: I don't understand why gemotypes are related to the features you mentioned. These may be related to the markers you use. Is there a wrong spelling? Correct and explain.
-Relate the phenological characteristics you examined with the performance in field performance. Indicate which primers (markers) are effective in selecting features. While genotype selection may be different for each study, for the reproducibility of your results, you should advise on which phenological characteristics and which ScoT markers selection can be made with in the early period, instead of genotype recommendation.
-For citation indications in the text, please carefully review the journal format and adapt it accordingly. Your representation doesn't fit anything.

Reviewer 1 ·

Basic reporting

no comment

Experimental design

no comment

Validity of the findings

no comment

Additional comments

The present study is well designed with appropriate analysis of obtained results. The authors have developed molecular markers associated with a set of quantitative phenotypic traits across 17 soybean genotypes, which might benefit the breeding the high-yielding soybean cultivars. In this regard, the submitted manuscript is of considerable significance and novelty.
1, Please further discuss the possible relationship between phenotypes at seedling stage and field performance as the seedling phenotypes might contribute to field performance, leading to correlation between multiple traits controlled by the same genetic loci.
2, Most of the data presented in the manuscript was properly analyzed. However, the number of samples from which the data were derived should be specified and the indication of statistical significance among groups was missing in the tables.
3, Though the language in the manuscript is generally clear and accurate, there are some places needing to be modified literally. Also, I suggest a thorough check of language to increase the comprehensibility of manuscript. My suggestions are as follow. (a) Remove the “content” in line 45; (b) Replace “so” by “Therefore” in line 50; (c) “The soybean genotypes (2n = 40) belong to the family Fabaceae” should be corrected because it is the species belongs to the family Fabaceae, but not the genotypes; (d) Line 55, “Sufficient genetic information regarding soybean traits is an essential step to making progress in soybean breeding programs” should be corrected.
4, The subtitles “The first experiment”, “The second experiment”, and “The third experiment” should be replaced by specified operations/experimental designing. Otherwise, it begets ambiguity as they might represent three biological replicates.
5, If possible, figures to show the 26 day-stage soybean plants and the field growth should be provided as supplementary materials.

Reviewer 2 ·

Basic reporting

no comment

Experimental design

no comment

Validity of the findings

no comment

Additional comments

The work is relevant and has application in the genetic improvement of plants, but needs some adjustments to the text; the introduction and objectives are adequate; the experiment was detailed and well conducted; the results are clear; the discussion is appropriate; and the conclusions are practical and respond to the objectives of the work. Therefore, I presented considerations in the original file with small corrections and suggestions.
This was just a summary of the comments made in the original file, justifying the opinion.

Annotated reviews are not available for download in order to protect the identity of reviewers who chose to remain anonymous.

---

## Round 0.2 · accepted · Accept

The corrections made are appropriate and your manuscript can be accepted. Congratulations